# Deepening ideas vs. exploring new ones: AI strategy effects in human-AI creative collaboration

**Kazuki Komura**[1,2]*, **Seiji Yamada**[1,2]

1 Department of Informatics, The Graduate University for Advanced Studies, SOKENDAI, Tokyo, Japan,
2 National Institute of Informatics, Tokyo, Japan

* kazukikomura@nii.ac.jp

## Abstract

As artificial intelligence (AI) increasingly participates in creative processes, designing effective human-AI collaboration is crucial. This study addresses a fundamental question: Should an AI partner prioritize deepening existing ideas (exploitation) or diversifying the creative space (exploration)? We investigated this through a controlled experiment with 148 participants, comparing two AI strategies based on March's exploration-exploitation framework. Using a turn-based brainstorming system on the topic of "how to increase café sales," we measured each AI strategy's impact on human trust and idea adoption. Contrary to traditional creativity research that emphasizes divergence, our findings show that the convergent deepening strategy significantly outperformed the diversification approach in both building user trust and encouraging idea adoption. We found that the AI's predictable behavior in the deepening condition was more easily understood by participants, leading to more effective collaboration.These results suggest that for effective human-AI co-creation, AI should act as a supportive, rather than competitive, partner that incrementally develops human-initiated concepts. This approach builds trust and increases the integration of AI's ideas. Our work thus contributes to the understanding of how AI can transition from tool-centric approaches to more collaborative partnerships that potentially enhance human creativity.

## Introduction

The complex challenges of modern society demand collaborative approaches that integrate diverse knowledge and cognitive abilities. In this context, the rapid development of artificial intelligence (AI) technology, particularly Large Language Models (LLMs), offers a groundbreaking opportunity to augment human creative capabilities and enable unprecedented forms of intellectual collaboration.

**Data availability statement:** All relevant data are within the manuscript and its Supporting information files.

**Funding:** JST, CREST (JPMJCR21D4), Japan. The funders (JST, CREST; JPMJCR21D4) had no role in study design, data collection and analysis, decision to publish, or preparation of the manuscript.

**Competing interests:** The authors have declared that no competing interests exist.

While traditional creative work has been conducted through human-to-human interaction, the emergence of LLMs has evolved AI from a mere information processing tool into a partner capable of dialogue and collaboration through language. Indeed, recent studies have shown that it is possible to generate ideas with LLMs adapted to an individual's creativity, and research exploring the division of roles and complementary relationships between AI and humans is actively being pursued [1–5].

To realize these technological possibilities, it is crucial to design human-AI collaboration not as a one-way support relationship but as a true collaborative partnership of mutual influence. As suggested by the concept of co-creative AI proposed by Kantosalo and Toivonen [6] and research on mutual adaptation processes by Seeber et al. [7], effective human-AI collaboration depends on complex design factors such as role allocation, interaction timing, and trust-building.

Nevertheless, research in this area is still in its nascent stages, with a lack of empirical findings on specific collaboration mechanisms for creative tasks like idea generation and brainstorming. Much of the current research on AI-assisted creativity focuses on a one-way approach where AI is utilized as a tool [8–11], leaving the development of a bidirectional collaborative model, where humans and AI participate as equal partners in the creative process, as a major research challenge.

Furthermore, improving the technical capabilities of AI alone is insufficient to achieve effective human-AI collaboration. Understanding the psychological and social aspects of how humans perceive and evaluate AI behavior is essential [12]. As Lee's theory of trust [13] indicates, trust in automated systems is vital for establishing appropriate reliance, and the calibration of trust is indispensable for the success of human-AI collaboration. Recent research has revealed that not only the performance of an AI system but also how it communicates its intentions, uncertainties, and limitations plays a crucial role in building trust [14].

This study addresses this challenge by posing a fundamental question regarding AI's behavior in creative collaboration: Should an AI partner prioritize the "deepening" or the "diversification" of thought? We empirically answer this question from two perspectives: human "trust" and "idea adoption."

Specifically, we conducted an empirical experiment that implemented two key idea-generation strategies in AI: a strategy that delves deeper into existing ideas and one that emphasizes the generation of new ideas. These approaches are based on March's [15] exploration-exploitation dilemma theory. This theory, which addresses the relationship between exploring new possibilities and exploiting existing certainties in organizational learning [16], serves as a key theoretical foundation in creativity research [17]. The deepening strategy embodies an "exploitative" approach, generating feasible solutions by qualitatively improving and specifying existing ideas. In contrast, the diffusion strategy embodies an "exploratory" approach, focusing on venturing into uncharted territories and expanding conceptual diversity.

Of particular importance is that the final decision to adopt an idea is left to the human. Therefore, it is crucial to determine which strategy influences the likelihood of idea adoption and to identify the characteristics of ideas that are more likely to be

selected. Furthermore, since proposals from an untrustworthy party may be rejected regardless of their quality [13], evaluating trust in the AI system itself is also a key subject of investigation.

To comprehensively examine these two aspects, we developed a grid-based UI system and conducted a between-participants experiment. We divided 148 participants into three groups: a deepening strategy (vertical), a diffusion strategy (horizontal), and a random control condition. In the experiment, humans and AI (GPT-4.1) alternately generated ideas on a 15x15 grid, and the effects of each strategy were measured across multiple dimensions: idea selection rate, trust in AI, and understanding of AI's behavior (Fig 1).

The contribution of this study lies in providing empirical data that supports concrete guidelines for interaction design in human-AI collaboration. This offers a critical empirical foundation for the paradigm shift from traditional tool-centric AI approaches to a true collaborative paradigm [1]. The importance of positioning AI not merely as an external tool but as a mutually adaptable collaborative partner in the integration of foundation models and tools is empirically supported by the results of this study. Furthermore, our findings align with recent research emphasizing the importance of a human-centered approach in the impact of generative AI on creativity and innovation.

The remaining sections summarize the current state of research on the challenges of group idea generation and human-AI collaboration frameworks to clarify our approach. We then describe the experiment and its results. Finally, we discuss the results and present our conclusions, limitations, and future work.

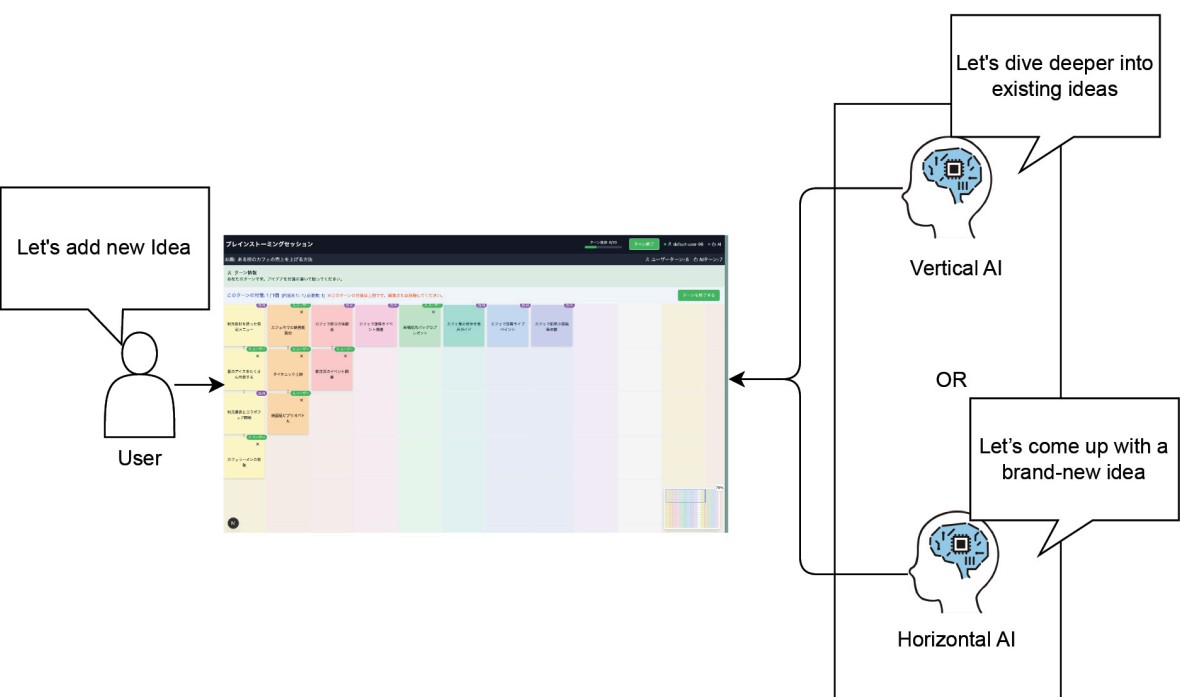

**Fig 1**. **Study overview: Human-AI collaborative brainstorming system.** Conceptual diagram illustrating the turn-based brainstorming system where users and AI agents collaborate on a shared grid interface. The system implements two distinct AI strategies: Vertical AI focuses on deepening existing ideas through incremental development, while Horizontal AI emphasizes exploring brand-new ideas to diversify the creative space. This study investigates which strategy is more effective for building trust and encouraging idea adoption in human-AI creative collaboration.

## Group creativity: Potential, challenges, and strategies

In theory, creative work in groups holds immense potential to produce outcomes that are difficult for individuals to achieve alone. The principles of brainstorming established by Osborn [18] and the concept of divergent thinking by Guilford [19] demonstrated the basic principle that "quantity breeds quality," clarifying that the quantitative production of diverse ideas is the foundation of creativity.

However, contrary to this theoretical promise, actual group creativity faces numerous constraints [20–23]. For instance, the "fixation effect," where one's thinking is constrained by the ideas of others, has been identified [23]. As a powerful strategic solution to this dilemma between "theoretical potential" and "practical constraints," March's [15] "exploration-exploitation dilemma" theory is positioned. It questions the balance between two conflicting learning approaches: "exploration," which pursues the discovery of new domains and diversity, and "exploitation," which aims at refining and improving the efficiency of existing knowledge. The diffusion strategy (exploration) adopted in this study is analogous to Mednick's [24] remote associates theory and seeks to ensure diversity by promoting the combination of heterogeneous knowledge, thereby overcoming constraints like the fixation effect. On the other hand, the deepening strategy (exploitation), similar to the "internalization" process in Nonaka and Takeuchi's [25] knowledge creation theory, aims to enhance the adoptability of ideas through the concretization and refinement of existing ideas.

Ultimately, as warned by Levinthal [26] in the "trap of myopic learning" and proposed by O'Reilly and Tushman [27] with the concept of the "ambidextrous organization," achieving a dynamic balance between exploration and exploitation, rather than leaning towards one, is the key to sustained creativity. While this study does not aim to achieve this balance through AI behavior, but rather to investigate human collaborative patterns under these two strategies, this theoretical background highlights the importance of our comparative verification of two different AI strategies.

## AI-assisted creativity and the collaborative paradigm shift

Current Large Language Models (LLMs) possess advanced idea generation capabilities and have been demonstrated to enhance human creative idea generation with appropriate support. This technological advancement suggests the potential for a shift from traditional tool-centric approaches to a true "co-creation" paradigm, where AI participates as an equal partner with humans in creativity support [6,28,29].

In fact, collaboration between LLMs and humans has been shown to improve both the process and outcomes of idea generation [1], realizing the ideal of human-centered creativity support proposed by Shneiderman [29]. A comprehensive review of creativity research in the HCI field by Gabriel et al. [30] indicates a trend towards placing more importance on collaborative creativity over individual creativity in the design of creativity support systems, supporting the significance of the human-AI collaboration that this study focuses on. The potential for "cognitive complementarity," where AI and humans mutually complement each other, suggests that AI collaboration can function as a collective intelligence [31] that surpasses the sum of individual abilities. From the perspective of Wiggins' [32] theory of exploratory spaces, the combination of AI's specificity and human abstraction could function as a mechanism to dynamically maintain the optimal balance of constraints and freedom essential for creativity.

## AI's exploration and exploitation strategies

In this new collaborative paradigm, research is underway to explore what strategies AI should adopt. The strategic importance of "exploration" and "exploitation," which this study focuses on, was clearly demonstrated by the empirical study of the CoQuest system by Liu et al. [33]. CoQuest revealed that a "breadth-first (exploration)" strategy, which explores diverse possibilities, and a "depth-first (exploitation)" strategy, which delves into specific topics, have different impacts on the creative process. This strategic contrast is deeply related to Guilford's [19] distinction between divergent and convergent thinking. Guilford [19] defined divergent thinking as "thinking that goes off in different directions" and associated it

with creativity due to its characteristic of generating diverse responses. Convergent thinking, on the other hand, focuses on finding a single correct answer to a well-defined problem.

This theoretical foundation is also closely related to a wide range of classic creativity theories, such as Finke's [34] Geneplore model, Mednick's [24] remote associates theory, and Smith's [35] constraint relaxation theory. Furthermore, excessive deepening risks generating repetitions and lowering evaluation scores. This latter finding is consistent with Martindale's [36] theory on the cycles of diffused and focused attention in creative cognition, suggesting that a single strategy is not always optimal. These findings are the direct motivation for the design of our study, which compares and verifies the effects of these two fundamental AI strategies from multiple perspectives.

## Trustworthiness and usability in AI collaboration

When designing AI strategies, human "trust" and "acceptability" are as important, if not more so, than technical performance. In the CoQuest study, trust-related issues emerged, with users expressing concerns about the originality of AI-generated ideas and hesitating to use them [33]. This indicates the difficulty humans have in appropriately trusting AI systems and correctly assessing their capabilities, suggesting that the calibration of an "appropriate level of trust" is essential [13]. A recent study by Chiou and Lee [14] emphasizes the need to understand trust as a dynamic adjustment process, captured from both responsiveness and resilience aspects.

This issue of trust can be understood in the broader context of human-automation interaction. For example, the finding in CoQuest that AI's processing delay gave users time for reflection and improved their sense of control can be understood as a design problem of "appropriate levels of automation" [37]. Furthermore, the finding that "transparency," which shows the AI's thought process, contributed to increased trust supports the claims of Explainable AI (XAI) research [38]. The Trust by Design framework by Merchan-Cruz et al. [39] demonstrates that trust is a dynamic, context-dependent element controlled by transparency, fairness, and user experience. Moreover, the observation that users spontaneously used various strategies (e.g., providing keywords, asking questions) with the AI suggests the importance of a mutual adaptation process where humans and AI dynamically adjust their roles [7].

## Research gap and positioning of this study

From the literature above, it is clear that the design of AI creativity support systems involves complex challenges such as the dynamic nature of strategy selection, the balance between exploration and exploitation, and trust-building. In response to these challenges, advanced collaborative models such as human-AI initiative-taking have been proposed [40]. Recent HCI research clearly shows a trend toward valuing collaborative creativity over individual creativity [41], emphasizing the need for co-creative systems that realize true partnerships. However, a fundamental barrier to effectively designing these advanced mechanisms is the lack of empirical evidence to answer the question, "On what criteria should strategies be dynamically switched?" For instance, even to implement a rule like "switch to another strategy when the user's trust begins to decline," we must first clarify the basic characteristics of each strategy—"which strategy is easier to build trust with and which is more likely to undermine it," and "which strategy is more likely to lead to idea adoption"—before the rule itself can be designed.

In essence, the future goal of achieving a "dynamic balance" can only become a concrete design challenge once a fundamental understanding of how the component polar strategies—exploration and exploitation—each affect human cognition and evaluation, namely trust and idea adoption, is established. Current research lacks this foundational empirical data, causing discussions on advanced collaborative models to lack specificity.

The study of the CoQuest system by Liu et al. [33] is a pioneering and important work that demonstrated the significance of "exploration (breadth-first)" and "exploitation (depth-first)" as AI strategies. However, CoQuest is a "batch generation model" where the AI generates and presents multiple RQ candidates at once based on user instructions. This

approach positions the AI as an advanced "search and suggestion tool." In contrast, our study employs a "turn-taking conversational model" where the human and AI contribute one idea at a time, alternately. This design positions the AI not as a mere tool but as an equal "dialogue partner," aiming for a more intimate and interactive collaborative relationship. In such a conversational context, it is not self-evident which strategy leads to trust or idea adoption.

Therefore, this study aims to build this essential foundation by returning to the fundamental principles of human-AI interaction proposed by Amershi et al. [12]. While this study may seem like a static and fundamental comparison of strategies, it is positioned as an essential first step toward the future goal of designing dynamic collaborative models. We will comparatively and empirically investigate the respective effects and side effects of the two basic strategies, "exploration" and "exploitation," from the perspectives of human trust and idea adoption.

### Research questions

To elucidate the impact of AI's collaborative strategies on human trust and idea adoption, this study establishes the following three research questions, each grounded in established theoretical frameworks.

First, we address the issue of trust in the AI partner. A system's predictability and consistency are known to be crucial for building user trust [13]. Our deepening (Vertical) strategy is designed to be predictable by consistently developing a specific idea, whereas the broadening (Horizontal) strategy may be harder for users to interpret as it shifts between topics. To clarify how these strategic differences impact user perception, we ask:

- **RQ1:** How do AI's collaborative strategies (deepening vs. broadening) affect trust ratings from users?

Second, we investigate how AI strategies affect overall idea selection patterns in collaborative brainstorming. While AI strategies directly influence the characteristics of AI-generated ideas, they may also have broader effects on the collaborative dynamic and overall selection patterns throughout the session [42,43]. The deepening strategy promotes focused development, while the broadening strategy encourages exploration across topics. Recent advances in human-AI collaboration have demonstrated varied impacts on creative outcomes [1,5]. To understand how these strategic differences impact the entire collaborative process, we ask:

- **RQ2:** How do different AI collaborative strategies affect overall idea selection patterns in human-AI brainstorming sessions?

Third, building on the first two questions, we conduct an exploratory analysis to understand the underlying mechanisms of idea selection. The choice to adopt an idea is likely influenced by more than just the AI's explicit strategy; it can depend on the idea's content, its visual placement, and its relationship with other ideas [44,45]. Research has shown that decision-making under uncertainty involves complex cognitive processes [46], while creative cognition relies on memory, attention, and cognitive control mechanisms [47]. To uncover these factors, we ask:

- **RQ3:** What factors influence idea selection? Specifically, to what extent do factors like semantic similarity, thematic relevance, and visual placement explain idea selection?

### Materials and method

This protocol was approved by the National Institute of Informatics Ethics Committee (Date: 2024-1-19, Approval No: R5-20-2). All research was conducted in accordance with the recommendations of the "Ethical Guidelines for Medical and Health Research Involving Human Subjects" provided by the Ministry of Education, Culture, Sports, Science and Technology and the Ministry of Health, Labour and Welfare. Informed consent was provided through an online form where participants selected one of the following options: "I indicate that I have read the information in the explanatory document

regarding participation in this study. I consent to participate in this study." All participants provided informed consent. They were then briefed on the experimental procedures.

## System overview and architecture

The developed system provides a turn-based collaborative brainstorming environment where a human and an AI (GPT-4.1) alternately generate and place ideas (Fig 2). This design is based on the concept of co-creative AI proposed by Kantosalo and Toivonen [6], positioning the AI as an active partner in the creative process rather than a mere auxiliary tool. It aims to obtain empirical findings on AI that contributes to the creative process on an equal footing with humans, going beyond the traditional Computer-Aided Creativity framework [28]. The AI understands all existing idea placements and content through a prompt to generate context-aware ideas.

The system employs a 15×15 grid-based interface as the space for placing ideas. The choice of this UI is rooted in March's [15] exploration-exploitation dilemma theory. To visually and intuitively map this theoretical framework, we defined horizontal expansion on the grid as "exploration" and vertical deepening as "exploitation," and implemented the AI's behavioral strategies on this two-dimensional space. This allows participants to implicitly or explicitly understand the AI's strategy through the spatial arrangement of ideas. The front end was built using Next.js, while the back end, responsible for saving placements and handling AI actions, was implemented using Express.

Each cell in the 15×15 grid space can contain a maximum of one idea (in a sticky note format). This constraint is intended to encourage participants to engage in strategic spatial placement and stimulate creative thinking [48]. The rules for idea placement are linked to the AI's strategy:

- **Vertical Placement (Exploitation):** "Deepens" an idea by adding a related idea within the same column where an existing idea is placed.

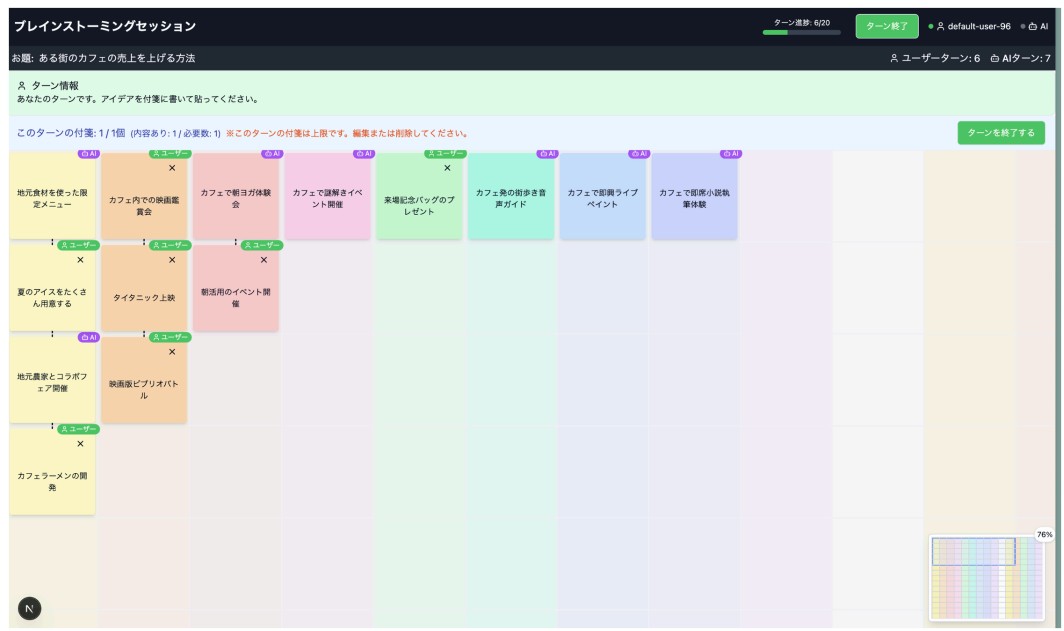

**Fig 2**. **System interface and idea selection process.** Left panel shows the main brainstorming interface with the 15x15 grid. Right panel displays the idea selection interface where participants evaluate generated ideas for adoption.

- **Horizontal Placement (Exploration):** "Expands" the scope by placing an idea in a new column, introducing a new perspective different from existing topics.

The interaction between the human and AI is governed by strict turn-taking control. This design promotes a dialogic and mutually adaptive relationship where one party responds after the other takes an action. In the experiment, the human and the AI each generated 20 ideas, creating a total of up to 40 ideas. To accommodate user errors, participants could freely delete or overwrite their own ideas. For each idea, we recorded the creator (user/AI), content, position on the grid (row, column), and creation time for later analysis. Users can also check the turn progress via a progress bar. Additionally, a minimap is positioned in the bottom-right corner of the system interface, allowing users to confirm their current location within the grid space.

### AI strategy implementation

Two AI behavioral strategies, Vertical (deepening) and Horizontal (broadening), were designed, with dedicated prompt engineering for each condition (Fig 3).

The Vertical strategy embodies March's [15] concept of 'exploitation.' The AI follows a deterministic algorithm to select a column for deepening: it scans the grid from left to right and chooses the first column that has not yet reached its maximum capacity. It then generates related ideas along that theme, thereby delving deeper into the topic. This process also corresponds to the 'internalization' process in Nonaka and Takeuchi's [25] knowledge creation theory, which transforms and deepens tacit knowledge into explicit knowledge.

In implementing this strategy, a crucial design decision was made regarding the maximum number of deepening steps. In the study of Liu et al.'s [33] CoQuest system, it was reported that continuing to generate ideas in depth led to increased repetitiveness and decreased evaluation scores beyond a certain depth (depth 9 or more). This suggests that the "fixation effect" [23], where the scope of thought narrows due to excessive influence from others' ideas, can also occur in the AI's generation process. Based on this finding, from a practical standpoint, our system was designed to limit the total number of ideas in a single column to a maximum of nine. This limit applies to the combined total of both human and AI-generated ideas in that column. Upon reaching nine, it automatically shifts its focus to exploring another column. This design prevents a decline in creativity due to excessive exploitation and encourages a moderate level of exploration.

Instructions to the AI are given through a prompt template containing the following elements:

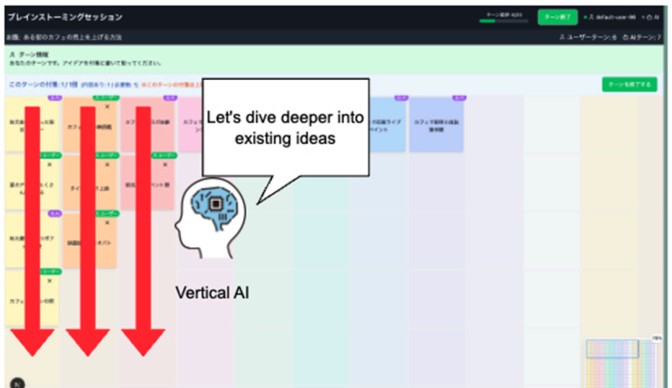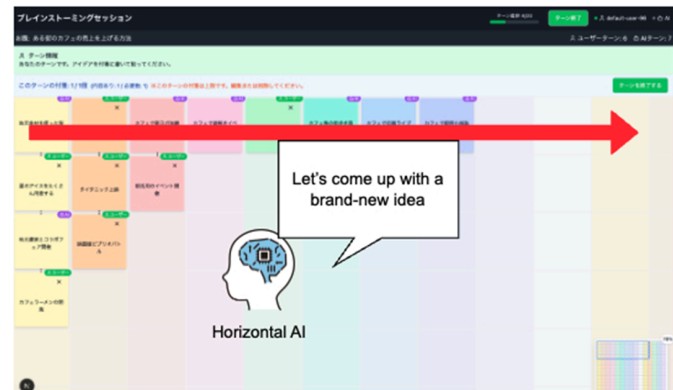

**Fig 3**. **AI strategy behavioral patterns.** Left panel shows the Vertical (deepening) strategy creating idea clusters in specific columns. Right panel displays the Horizontal (broadening) strategy spreading ideas across different columns to maximize diversity.

- **Task Definition:** Your task is "Vertical Exploration." This means deepening and developing a specific existing idea.
- **Contextual Information:** The session's theme, the content and position of all existing ideas, the parent idea to be deepened, and its column number.
- **Behavioral Constraints:** You must place the idea in the specified column (targetColumn). The content must be a derivation, development, or deepening of the parent idea. The idea should be concise, around 15 characters. The output must follow the specified JSON format (["idea": "content", "position": "row": Y, "col": X]).

This prompt design ensures the AI performs consistent deepening on a given topic, generating predictable and understandable behavior. All AI responses were generated using the OpenAI API with the following parameters: model specified by `OPENAI_MODEL`, temperature set to 0.0 for consistent outputs, `max_tokens` limited to 400, $n = 1$ for single response generation, and `response_format` configured as JSON object for structured output. The complete prompt templates and technical implementation details are provided in the Supporting Information (see S1 File).

The specific placement algorithm for the Vertical strategy is as follows:

**Algorithm 1 Vertical (deepening) strategy placement algorithm.**

```
 1: Initialize:
 2: Grid dimensions: 15 × 15, Maximum ideas per column: MAX_DEPTH = 9
 3: Current grid state: grid[row][col]
 4: Input: Current grid state, existing ideas
 5: Output: Target position (target_row, target_col)
 6: target_col ← −1
 7: target_row ← −1
 8: for col = 0 to 14 do
 9:   depth ← count of ideas in column col
10:   if depth < MAX_DEPTH then
11:     target_col ← col
12:     target_row ← depth
13:     break
14:   end if
15: end for
16: return   (target_row, target_col)
```

The Horizontal strategy embodies March's [15] concept of 'exploration.' The AI preferentially selects new columns with no existing ideas, or columns with few ideas, to generate diverse ideas with low relevance to existing topics. This approach, as indicated by Mednick's [24] remote association theory, aims to foster creativity by connecting semantically distant concepts.

This strategy, similar to the breadth-first approach shown by Liu et al. [33], encourages a broad exploration of the potential idea space. Presenting diverse ideas, especially in the initial phase, is known to have a significant impact on the subsequent human thought process [49]. However, an overly large exploration space can increase cognitive load. Brainstorming research has reported that individuals can reach a state of "idea generation depletion" where they can no longer produce new ideas, indicating cognitive limits. For this reason, our design shifts to an exploratory idea generation based on derived ideas once the limited grid exploration is complete.

- **Task Definition:** Your task is "Horizontal Exploration." This means proposing ideas with a completely new direction, different from existing ideas.
- **Contextual Information:** The session's theme, the content and position of all existing ideas, and the target position (row and column) for placing the idea.

- **Behavioral Constraints:** You must place the idea at the specified position (targetPositions). If placing in row 0 (the first idea), generate an idea from a completely new perspective. If placing in row 1 or later, generate an idea derived from the idea in row 0 of the same column (the parent idea). The idea should be concise, around 15 characters. The output must follow the specified JSON format.

This prompt enables the AI to systematically traverse the idea space, providing diverse perspectives and strongly supporting the divergent phase of brainstorming. The complete prompt templates and API configuration details are provided in the Supporting Information (see S1 File).

The specific placement algorithm for the Horizontal strategy is as follows:

**Algorithm 2** Horizontal (broadening) strategy placement algorithm

```
 1: Initialize:
 2: Grid dimensions: 15 × 15
 3: Current grid state: grid[row][col]
 4: Input: Current grid state, existing ideas
 5: Output: Target position (target_row, target_col)
 6: target_row ← −1
 7: target_col ← −1
 8: for row = 0 to 14 do
 9:   filled_positions ← count of filled positions in row row
10:   if filled_positions < 15 then
11:     target_row ← row
12:     for col = 0 to 14 do
13:       if grid[row][col] is empty then
14:         target_col ← col
15:         break
16:       end if
17:     end for
18:     break
19:   end if
20: end for
21: if target_row = 0 then
22:   Generate completely new perspective idea
23: else
24:   if grid[0][target_col] exists then
25:     Generate idea derived from grid[0][target_col] (parent idea)
26:   else
27:     Generate completely new perspective idea
28:   end if
29: end if
30: return   (target_row, target_col)
```

## Study design and conditions

This study conducted a one-factor, three-level between-participants experiment. Specifically, the following three experimental conditions were set:

- **Vertical exploration (deepening) condition:** Based on March's (1991) exploitation strategy, the AI decides on an existing idea (a certain column) and delves deeper into related ideas.
- **Horizontal exploration (broadening) condition:** Based on March's (1991) exploration strategy, the AI explores ideas that are different from or have low relevance to existing ideas by targeting new columns or columns with lower hierarchy.

- **Random condition:** A control group where the AI randomly selects between Vertical and Horizontal strategies.

Based on a sample size calculation using G*Power by Faul et al. (effect size $f = 0.25$, power = 0.8, significance level $\alpha = .05$), we recruited 165 participants between May 24,2025 and May 25, 2025 via Yahoo! Crowdsourcing, with a compensation of 70 JPY each. After excluding those who experienced GPT API errors, 148 participants were included in the final analysis. The participants consisted of 115 males (77.7%), 32 females (21.6%), and one other (0.7%). The mean age was 47.93 years (SD = 10.56, median = 49.0). Table 1 shows the distribution of participants across the experimental conditions. No prior survey was conducted on participants' proficiency with AI or their experience with creative tasks, as this study aimed to investigate trust in and actions toward AI regardless of participants' expertise.

## Experimental procedure

The experiment proceeded in a sequence of a pre-briefing for participants, a practice session, the main session, evaluation of the generated ideas, and a post-session questionnaire (Fig 4). First, participants were given a briefing that included video instructions. These instructions explained how to operate the system and, importantly, provided explicit guidance with examples on how to strategically place ideas on the grid. Following this briefing, a practice session was provided for participants to get accustomed to the interface through an actual brainstorming experience. This was intended to provide a "safe learning environment," which is considered important in Edmondson's [51] theory of psychological safety. Detailed user interface instructions and experimental materials are provided in the Supporting Information (see S2 File).

In the subsequent main session, a 15-minute brainstorming session was conducted on the theme "How to increase sales for a city cafe." In this session, the participant and the AI alternately generated ideas, producing a maximum total of 40 ideas. The session duration was set considering the appropriate length recommended in research on group creativity by Paulus and Nijstad [52].

After the session, participants moved to an idea evaluation phase where they selected one or more ideas they would "actually want to adopt" from all the generated ideas. To avoid potential bias from grid positioning, these evaluation sessions were conducted independently of the grid layout, presenting ideas in a neutral format without spatial context.

**Table 1**. **Participant distribution across experimental conditions.** Number of participants in each AI strategy condition after excluding those with technical issues.

| Condition | Participants (N) |
|---|---|
| Horizontal (Broadening) | 50 |
| Vertical (Deepening) | 52 |
| Random (Control) | 46 |
| Total | 148 |

Note: Participant count is based on standard sample size guidelines [50] to ensure adequate statistical power.

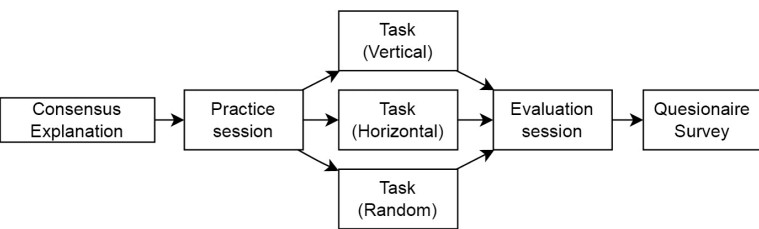

**Fig 4**. **Experimental procedure flow.** Diagram showing the complete experimental procedure from participant briefing through practice session, main brainstorming session, idea evaluation, to post-session questionnaire.

Finally, participants were asked to answer a questionnaire created with reference to Lee and See's [13] research on trust in automated systems and Amabile's [53] creativity assessment methods. The questionnaire assessed participants' perception of the AI's behavior patterns, their trust in the AI, and the quality of the collaborative experience.

## Evaluation metrics and analysis

In this study, we evaluated the experimental results from a multi-faceted approach, including quantitative metrics, qualitative metrics, behavioral analysis metrics, and text analysis. Our analytical approach consisted of three main components: (1) descriptive analysis of idea selection patterns and trust ratings across experimental conditions, (2) between-group comparisons using ANOVA and Kruskal-Wallis tests for normally and non-normally distributed data respectively, and (3) mixed-effects logistic regression modeling to identify factors influencing idea selection while controlling for participant-level variation.

To evaluate the quantity and selection tendency of ideas, several quantitative metrics were measured. As an indicator of idea fluency, we used the total number of ideas selected by the participant. As a breakdown, we recorded the number of selected ideas generated by the participant and the number of selected ideas generated by the AI.

To measure participants' subjective evaluations, we included questions about their understanding of the AI's behavior, their trust, and the collaborative experience. Regarding the understanding of the AI's behavior patterns, we asked whether the AI's behavior seemed to be "deepening" ideas, "generating" new ideas, or if it was "unclear."

Trust in the AI was measured based on Lee and See's [13] Multidimensional Measure of Trust (MDMT), from the aspects of reliability, capability, ethicality, and sincerity. From these subscales, we calculated competence trust and moral trust.

Furthermore, to measure perceptual evaluation of the AI system, we adopted the Godspeed Questionnaire Series (GQS) developed by Bartneck et al. [54]. The GQS is a scale that evaluates robots or AI systems on five dimensions: Anthropomorphism, Animacy, Likeability, Perceived Intelligence, and Perceived Safety. Each dimension was measured using a 5-point semantic differential scale, quantifying how human-like, animate, likeable, intelligent, and safe the AI system was perceived to be.

To evaluate the semantic content of the generated ideas, we conducted analysis through text mining. Following approaches that use semantic similarity measures for creativity evaluation [45], we converted each idea into an embedding vector using text embedding models, enabling quantitative analysis of semantic relationships between ideas. From this vector data, we calculated the following metrics: First, "thematic similarity," indicating the semantic similarity to the main theme of the brainstorming. Second, "average similarity to others," defined as the average similarity to other ideas within the same session. Third, "maximum similarity to others," the similarity to the most similar other idea. Fourth, "minimum similarity to others," the similarity to the least similar idea. Fifth, "standard deviation of similarity to others," indicating the variability of similarity to other ideas. These metrics are an attempt to quantify the processes of "blind variation" and "selective retention" in Campbell's [55] theory of creativity.

To comprehensively analyze the factors influencing idea selection, we constructed a mixed-effects logistic regression model. In this model, idea selection (a binary variable) was the dependent variable, and the AI condition (Vertical, Horizontal, Random), idea creator (AI vs. human), grid position (row, col), idea content length, and the aforementioned similarity metrics were included as fixed effects. The participant (sessionId) was set as a random effect to control for individual differences among participants. To address potential statistical concerns, all continuous variables were standardized (z-score normalization) to facilitate interpretation and reduce scale-related artifacts. Multicollinearity was assessed using Variance Inflation Factor (VIF) analysis, and complete pairwise comparisons between all strategy conditions were conducted using the emmeans package to ensure comprehensive strategy evaluation.

## Results

### User perception of the system

Measuring participants' perception of the AI's behavior patterns revealed a significant difference among the strategies ($\chi^2(4, N = 148) = 23.09, p = .0001$, Cramer's $V = .279$). In the Vertical condition, 61.5% (32/52) of participants perceived the AI's behavior as "deepening." In the Horizontal condition, 54.0% (27/50) perceived it as "generating new ideas," and in the Random condition, 45.7% (21/46) also perceived it as "generating."

The proportion of participants who responded "unclear" was 7.7% (4/52) in the Vertical condition, 28.0% (14/50) in the Horizontal condition, and 23.9% (11/46) in the Random condition.

In the trust evaluation using MDMT scale [13] , the following results were obtained:

**MDMT trust scale results.** As shown in Table 2 and Fig 5, the Vertical condition demonstrated superior performance across multiple trust dimensions compared to the Horizontal and Random conditions. Most notably, Competence Trust showed a significant difference ($F(2, 145) = 5.07, p = 0.007, \eta^2 = 0.065$), with the Vertical condition achieving a mean score of 4.79 compared to 4.04 for Horizontal and 4.16 for Random. Post-hoc tests revealed significant differences between Vertical and Random conditions ($p = 0.010$) and between Vertical and Horizontal conditions ($p = 0.040$). Similarly, Moral Trust showed significant differences ($F(2, 145) = 4.12, p = 0.018, \eta^2 = 0.054$), with post-hoc tests confirming significant differences between Vertical and Random ($p = 0.048$) and between Vertical and Horizontal ($p = 0.032$).

**Godspeed Questionnaire (GQS) results.** Table 3 presents the results of the Godspeed Questionnaire evaluation across the three experimental conditions. Among the five dimensions measured, Perceived Safety showed a marginally significant trend ($F(2, 145) = 3.06, p = 0.050, \eta^2 = 0.041$). The Vertical condition achieved the highest score (M = 3.93, SD = 0.73), outperforming both the Horizontal (M = 3.64, SD = 0.65) and Random (M = 3.61, SD = 0.77) conditions. However, post-hoc analysis did not reveal significant pairwise differences in the Tukey HSD test. In contrast, no significant differences were observed for the remaining four dimensions: Anthropomorphism ($F(2, 145) = 0.76, p = 0.469$), Animacy ($F(2, 145) = 1.08, p = 0.341$), Likeability ($F(2, 145) = 0.65, p = 0.525$), and Perceived Intelligence ($F(2, 145) = 1.38, p = 0.256$).

### Idea acceptance

**AI idea selection results.** As shown in Table 4 and Fig 6, significant differences were confirmed in the number of AI ideas selected across the three experimental conditions. The Vertical condition demonstrated the highest performance with an average of 6.98 selected AI ideas, significantly outperforming the Horizontal condition (5.22 selected ideas). The Random condition showed intermediate performance (6.20 selected ideas). A Kruskal-Wallis H-test confirmed significant differences ($H(2) = 6.71, p = 0.035$), with a small to medium effect size ($\eta^2 = 0.033$).

**Analysis of grid position usage via visualization.** To analyze the spatial distribution patterns of ideas across all 148 sessions, we created heatmaps of the grid positions. Visualizing all 5,848 idea placements revealed a tendency for

**Table 2**. MDMT trust scale results by AI strategy condition.

| Trust Dimension | Vertical | Horizontal | Random | F-statistic | p-value | $\eta^2$ |
|---|---|---|---|---|---|---|
| Reliability | 4.76 (1.13) | 4.08 (1.37) | 4.09 (1.30) | 4.79 | 0.010* | 0.062 |
| Capability | 4.81 (1.26) | 4.00 (1.52) | 4.22 (1.31) | 4.76 | 0.010* | 0.062 |
| Ethicality | 4.71 (1.22) | 4.21 (1.35) | 4.11 (1.30) | 3.12 | 0.047* | 0.041 |
| Sincerity | 4.92 (1.08) | 4.32 (1.27) | 4.31 (1.05) | 4.82 | 0.009** | 0.062 |
| Competence Trust | 4.79 (1.11) | 4.04 (1.42) | 4.16 (1.26) | 5.07 | 0.007** | 0.065 |
| Moral Trust | 4.82 (1.12) | 4.26 (1.28) | 4.21 (1.11) | 4.12 | 0.018* | 0.054 |

Note: *p < 0.05, **p < 0.01, †p < 0.10 (trend). ANOVA F-tests were conducted to compare means across the three conditions. Effect sizes are reported as partial eta-squared.

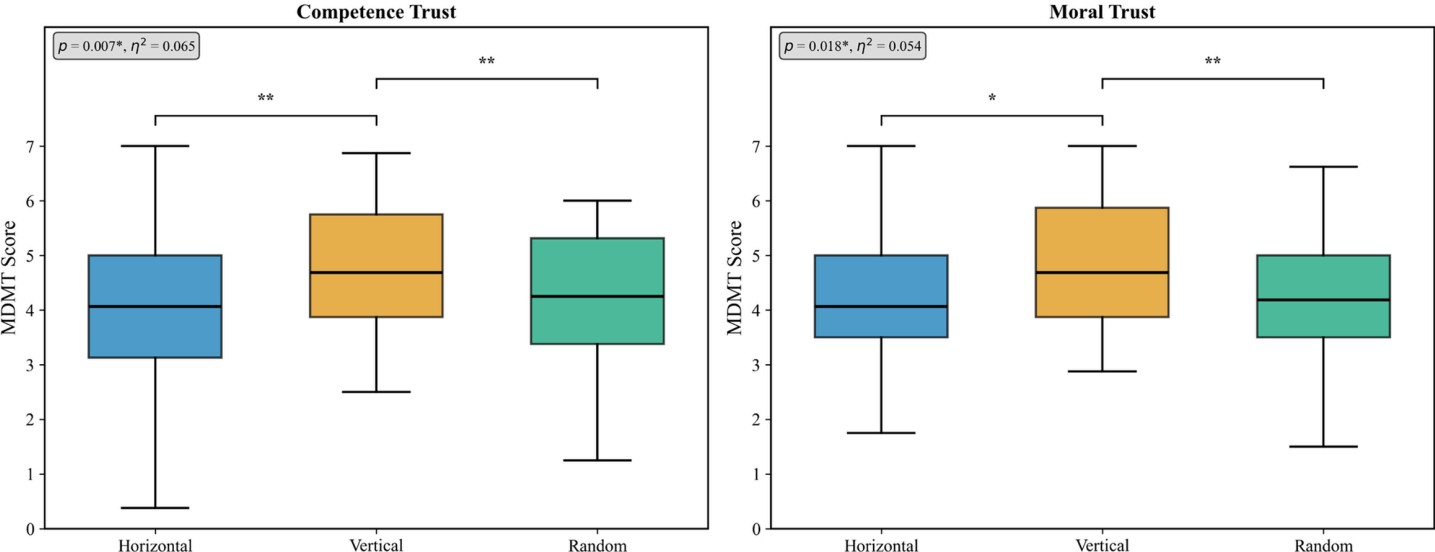

**Fig 5**. **Trust evaluation results.** MDMT trust scale results showing significantly higher trust ratings for the Vertical strategy across multiple dimensions including competence trust and moral trust.

**Table 3**. **Godspeed questionnaire results by AI strategy condition.**

| GQS Dimension | Vertical | Horizontal | Random | F-statistic | p-value | $\eta^2$ |
|---|---|---|---|---|---|---|
| Anthropomorphism | 2.93 (0.81) | 2.75 (0.91) | 2.93 (0.89) | 0.76 | 0.469 | 0.010 |
| Animacy | 3.68 (0.78) | 3.50 (0.78) | 3.45 (0.88) | 1.08 | 0.341 | 0.015 |
| Likeability | 3.57 (0.69) | 3.43 (0.64) | 3.45 (0.67) | 0.65 | 0.525 | 0.009 |
| Perceived Intelligence | 3.70 (0.81) | 3.45 (0.77) | 3.57 (0.75) | 1.38 | 0.256 | 0.019 |
| Perceived Safety | 3.93 (0.73) | 3.64 (0.65) | 3.61 (0.77) | 3.06 | 0.050* | 0.041 |

Note: *$p < 0.05$. ANOVA F-tests were conducted to compare means across the three conditions. Effect sizes are reported as partial eta-squared.

**Table 4**. **Comprehensive idea selection analysis by strategy condition.** Number of ideas selected across different categories, showing AI strategies primarily affect AI-generated idea adoption rather than overall collaborative productivity.

| Selection Category | Vertical | Horizontal | Random | H-statistic | p-value |
|---|---|---|---|---|---|
| AI Ideas Selected | 6.98 (5.13) | 5.22 (5.43) | 6.20 (4.89) | 6.71 | 0.035* |
| Human Ideas Selected | 6.25 (4.85) | 6.22 (5.65) | 4.85 (5.13) | 3.44 | 0.179 |
| Total Ideas Selected | 13.23 (8.77) | 11.44 (9.20) | 11.04 (9.42) | 3.55 | 0.169 |

Note: Values shown as mean (standard deviation). *$p < 0.05$, n.s. = not significant. Kruskal-Wallis H-test was used due to non-normal distribution. Post-hoc analysis for AI Ideas: Horizontal vs Vertical, $p = 0.037$*; other pairwise comparisons non-significant. Human and Total selection showed no significant pairwise differences.

ideas to be concentrated in the upper and left portions of the grid. A heatmap of only the selected ideas showed this trend more prominently, with selected ideas clearly concentrated in the upper-left region of the grid (rows 0-2, columns 0-3, see Fig 7). Analysis of selection rates by position observed a gradient, with rates being higher closer to the top-left corner of the grid and decreasing towards the bottom-right.

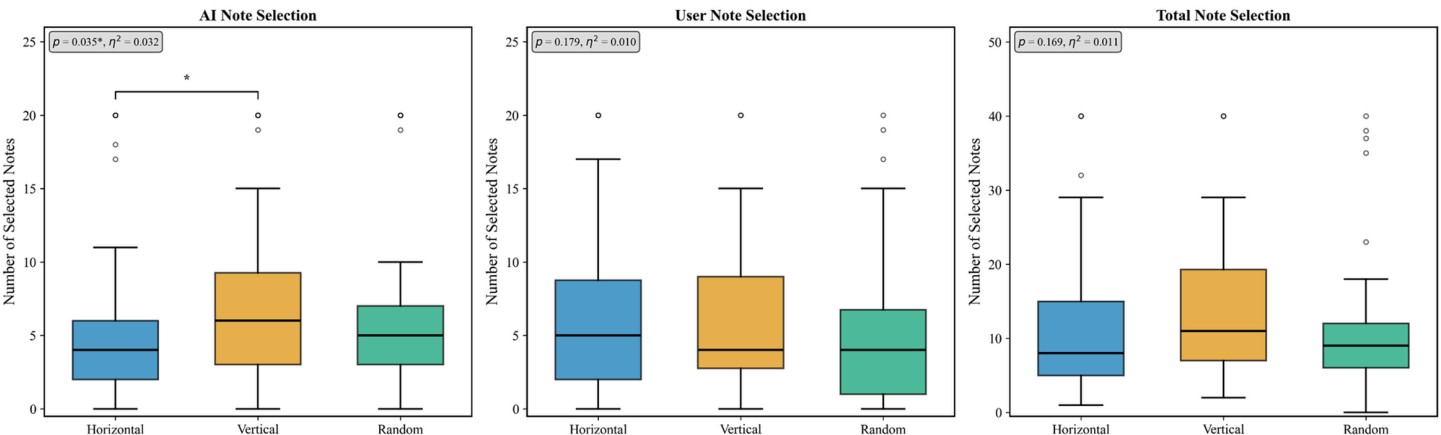

**Fig 6**. **AI idea selection analysis.** Plots showing the distribution of the number of AI ideas selected across experimental conditions. The Vertical strategy demonstrates significantly higher selection counts with reduced variability.

**t-SNE visualization of the latent space.** To analyze the semantic similarity of ideas, we performed dimensionality reduction on the embedding vectors using t-SNE (t-distributed Stochastic Neighbor Embedding) for 2D visualization. Fig 8 presents a two-panel analysis comparing the distribution of ideas between horizontal and vertical AI strategies.

Panel A shows the distribution by AI strategy, where ideas from the vertical condition appear in relatively dense clusters while ideas from the horizontal condition are more widely dispersed across the latent space. Panel B illustrates the

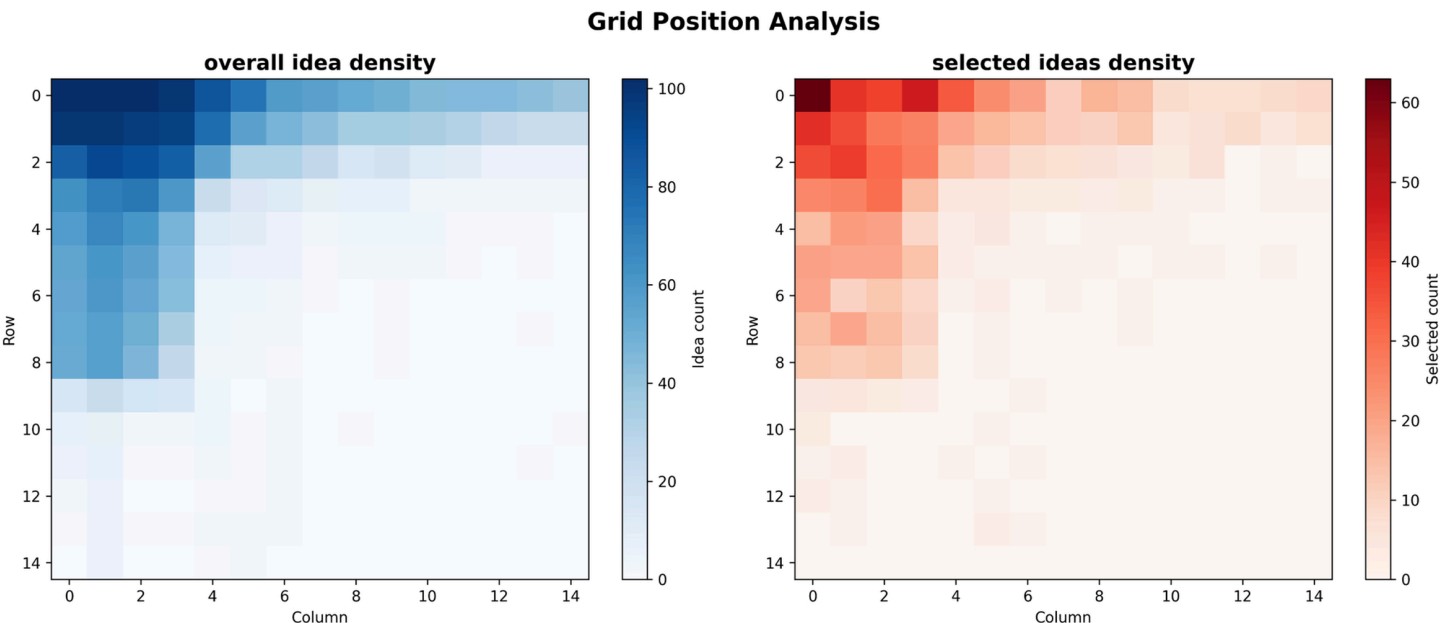

**Fig 7**. **Grid position analysis.** Heatmaps showing spatial distribution of ideas and selection rates across the 15x15 grid. Demonstrates the positional bias with higher selection rates in the upper-left regions.

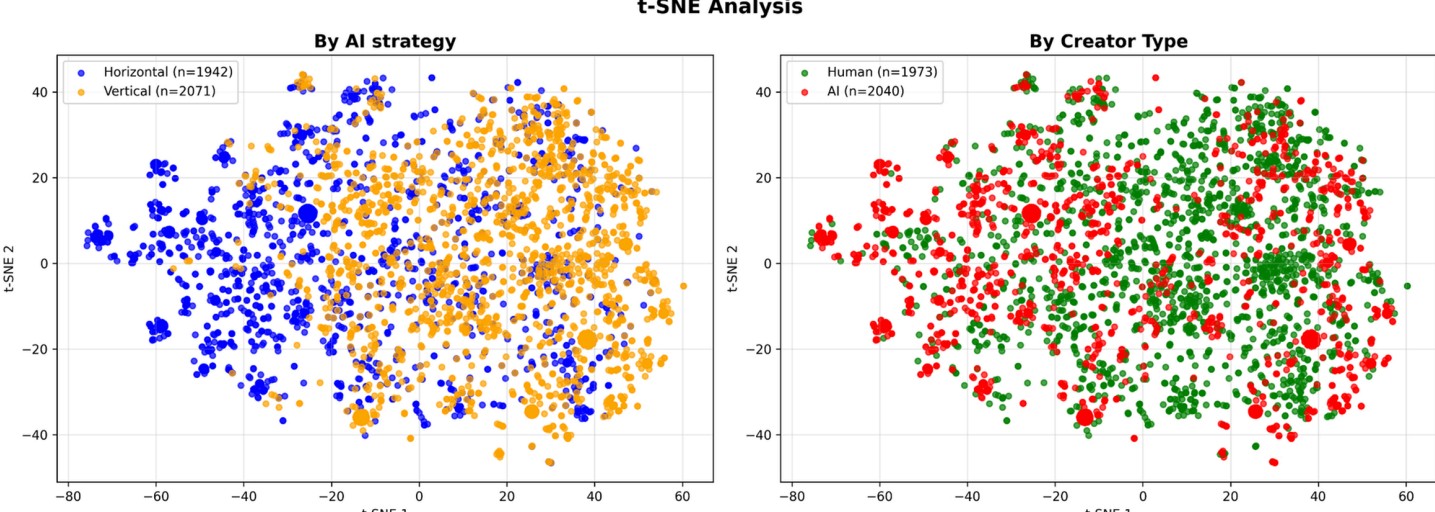

**Fig 8. t-SNE embedding analysis of brainstorming ideas.** Two-panel t-SNE visualization showing: (A) Distribution by AI strategy with distinct clustering patterns between horizontal and vertical conditions, and (B) Distribution by creator type revealing overlapping but distinct regions for human and AI ideas.

distribution by creator type, revealing that ideas created by humans and those generated by the AI partially overlap in the latent space but also tend to concentrate in distinct areas, with human ideas distributed across more diverse regions while AI ideas show relatively concentrated distributions.

**Mixed-effects logistic regression results.** A mixed-effects logistic regression model was constructed to analyze the 5,848 ideas from 148 participants, achieving good discriminative performance (AUC = 0.799). All continuous variables were standardized (z-score normalization) to ensure interpretable odds ratios and facilitate comparison across different scales.

**Multicollinearity assessment:** VIF analysis revealed minimal concerns, with all variables showing VIF <5 except similarity standard deviation (VIF = 5.10, indicating mild collinearity). This validates the statistical approach and model stability.

As presented in Table 5, several factors significantly influenced idea selection in the final model that includes position-content interaction terms. The analysis revealed a moderate positive effect of average similarity to other ideas (OR = 1.37, $p < 0.001$), indicating that ideas semantically similar to existing ones were more likely to be selected. Positional effects were highly significant, with ideas positioned higher (OR = 0.716 per SD for grid row) and further left (OR = 0.664 per SD for grid column) showing increased selection probability. Content length had a modest positive effect (OR = 1.17, $p < 0.001$). Notably, most position-content interactions were non-significant, validating the interpretation of main effects.

**Position-content independence testing:** To address potential confounding between positional and content effects, we systematically tested for interactions between position and content variables in our mixed-effects logistic regression model. The analysis included all possible interaction terms between grid coordinates (row, column) and content features (average similarity, maximum similarity, minimum similarity, similarity standard deviation, theme similarity, and content length).

The comprehensive interaction analysis revealed limited significant interactions: only 2 out of 12 tested interactions were statistically significant at $p < 0.05$. Specifically, grid row × minimum similarity ($p < 0.001$) and grid column × minimum

**Table 5. Factors significantly affecting idea selection.** Mixed-effects logistic regression results with standardized variables and Creator control variable showing main effects on idea selection (N = 5,848 ideas from 148 participants).

| Factor | Odds Ratio | 95% CI | p-value |
|---|---|---|---|
| *Positional Effects (per SD)* | | | |
| Grid Row (Standardized) | 0.719 | 0.661-0.782 | < 0.001*** |
| Grid Column (Standardized) | 0.664 | 0.606-0.728 | < 0.001*** |
| *Content Characteristics (per SD)* | | | |
| Content Length (Standardized) | 1.15 | 1.07-1.24 | < 0.001*** |
| *Similarity Metrics (per SD)* | | | |
| Average Similarity (Standardized) | 1.35 | 1.16-1.58 | < 0.001*** |
| Maximum Similarity (Standardized) | 1.41 | 1.23-1.63 | < 0.001*** |
| Minimum Similarity (Standardized) | 0.824 | 0.737-0.922 | < 0.001*** |
| SD of Similarity (Standardized) | 0.407 | 0.323-0.513 | < 0.001*** |
| Theme Similarity (Standardized) | 1.04 | 0.947-1.13 | 0.441 |
| *AI Strategy Condition* | | | |
| Vertical vs. Random | 1.13 | 0.658-1.95 | 0.658 |
| Horizontal vs. Random | 1.04 | 0.600-1.81 | 0.882 |
| *Creator Effect* | | | |
| AI vs. Human | 1.01 | 0.864-1.18 | 0.945 |

Note: Model includes Creator variable and standardized continuous variables (z-score). VIF analysis detected minor multicollinearity (std_similarity VIF = 5.10). All strategy comparisons are non-significant when controlling for Creator effects.

similarity ($p = 0.041$) showed significant interactions. All other position-content interactions, including those with average similarity, maximum similarity, similarity standard deviation, theme similarity, and content length, were non-significant ($p > 0. 05$).

These results indicate that position and content effects operate largely independently, with only minimal interaction confined to minimum similarity effects. The interaction effects are substantively small compared to the main effects, and the vast majority of tested interactions (10/12) were non-significant, supporting the overall conclusion that positional and content factors represent distinct pathways through which AI strategies influence idea selection. Table 6 presents the complete interaction analysis results.

**Table 6. Comprehensive position-content interaction analysis.** Mixed-effects logistic regression results testing for interactions between positional and content effects (N = 5,848 ideas from 148 participants).

| Interaction Term | Estimate | z-value | p-value |
|---|---|---|---|
| Grid Row × Average Similarity | 0.044 | 0.58 | 0.562 |
| Grid Column × Average Similarity | −0.005 | −0.07 | 0.941 |
| Grid Row × Maximum Similarity | −0.026 | −0.33 | 0.739 |
| Grid Column × Maximum Similarity | −0.055 | −0.69 | 0.491 |
| Grid Row × Minimum Similarity | −0.193 | −3.42 | < 0.001*** |
| Grid Column × Minimum Similarity | −0.114 | −2.05 | 0.041* |
| Grid Row × Similarity SD | −0.093 | −0.78 | 0.438 |
| Grid Column × Similarity SD | −0.131 | −1.13 | 0.256 |
| Grid Row × Theme Similarity | 0.026 | 0.52 | 0.606 |
| Grid Column × Theme Similarity | −0.004 | −0.08 | 0.934 |
| Grid Row × Content Length | −0.009 | −0.23 | 0.819 |
| Grid Column × Content Length | −0.075 | −1.83 | 0.068 |

Note: *p < 0.05, **p < 0.01, ***p < 0.001. Only 2 out of 12 interactions were statistically significant, indicating largely independent position and content effects. Significant interactions are confined to minimum similarity, with effect sizes substantially smaller than main effects.

## Discussion

This study empirically investigated and compared two AI strategies, deepening priority (Vertical) and broadening priority (Horizontal), for the specific interaction design problem in creative ideation within human-AI groups. The following discusses the findings and their implications for the three established research questions (RQ1-RQ3).

### RQ1: The impact of AI strategy on trust evaluation

The experimental results provided a clear answer to RQ1: "How do AI's collaborative strategies (deepening strategy vs. broadening strategy) affect trust ratings from users?" The deepening strategy (Vertical) was found to receive significantly higher ratings on multiple trust indicators compared to the broadening strategy (Horizontal) and the random condition. Specifically, significant differences were observed for Competence Trust ($F(2, 145) = 5.07, p = 0.007$) and Moral Trust ($F(2, 145) = 4.12, p = 0.018$).

The fact that 61.5% of participants accurately recognized the deepening strategy suggests that the predictability of the behavior pattern contributed to trust-building. In Lee and See's [13] theory of trust in automated systems, predictability is identified as one of the fundamental components of trustworthiness, specifically contributing to the "reliability" dimension of trust through consistent and comprehensible behavior patterns.

Our findings provide empirical support for this theoretical framework in human-AI creative collaboration. The significantly higher recognition rate of the Vertical strategy (61.5% vs. 28.0% "unclear" responses for Horizontal) suggests that behavioral transparency directly influences trust formation. While our study design does not establish direct causality between recognition and trust, the concurrent observation of high recognition rates and superior trust scores in the Vertical condition aligns with Lee and See's [13] predictability-trust hypothesis. The consistency of this pattern across multiple trust dimensions (Competence Trust, Moral Trust, and Perceived Safety) strengthens the theoretical foundation that predictable AI behavior facilitates appropriate trust calibration in collaborative settings.

In contrast, 28.0% of participants in the broadening strategy condition rated the AI's behavior as "unclear," which may have led to lower trust ratings. When the AI's behavior is difficult to understand, participants cannot adequately evaluate its performance, which in turn hinders the building of a trust relationship. This finding is consistent with research on automation transparency [38], which emphasizes that system comprehensibility is essential for trust formation.

### RQ2: The effect of strategy on overall idea selection patterns

The data provided a clear answer to RQ2: "How do different AI collaborative strategies affect overall idea selection patterns in human-AI brainstorming sessions?" Comprehensive analysis of idea selection patterns revealed that AI strategies primarily influenced the adoption of AI-generated ideas, with more limited effects on overall collaborative patterns.

For AI-generated ideas specifically, the deepening strategy significantly outperformed other strategies. The number of AI-generated ideas selected was significantly higher for the deepening strategy (6.98 ideas) than for the broadening strategy (5.22 ideas) ($H(2) = 6.71, p = 0.035$). This difference of 1.76 ideas per session represents a 34% improvement and is significant from both statistical and practical perspectives. The Random condition showed intermediate performance (6.20 selected ideas).

Importantly, this advantage did not extend to human-generated ideas, where no significant differences were observed across strategies (Deepening: 6.25 ideas, Broadening: 6.22 ideas, Random: 4.85 ideas; $H(2) = 3.44, p = 0.179$). Similarly, total idea selection showed no significant differences between strategies (Deepening: 13.23 ideas, Broadening: 11.44 ideas, Random: 11.04 ideas; $H(2) = 3.55, p = 0.169$), indicating that the effects were specific to AI contributions rather than overall collaborative productivity.

This pattern suggests that AI strategies achieve their effectiveness by generating ideas that align with human cognitive preferences for contextually relevant contributions, as will be further explained through the mechanism analysis in RQ3.

**RQ3: An exploratory analysis of factors influencing idea selection**

The mixed-effects logistic regression analysis for RQ3, "What factors influence idea selection? Specifically, to what extent do factors like semantic similarity, thematic relevance, and visual placement explain idea selection?", identified specific factors that influence idea selection across all ideas in the brainstorming sessions. This comprehensive analysis examined all ideas (both human and AI-generated) to understand the underlying mechanisms that explain the patterns observed in RQ2.

We conducted additional robustness analyses including variable standardization, multicollinearity assessment, and complete pairwise strategy comparisons. As presented in Table 7, the standardized analysis confirmed that AI strategy conditions showed no significant main effects (Vertical vs. Random: OR = 1.13, p = 0.658; Horizontal vs. Random: OR = 1.04, p = 0.882; Vertical vs. Horizontal: OR = 1.09, p = 0.765) when examining all ideas collectively. Variance Inflation Factor analysis revealed only minor multicollinearity concerns (one variable with VIF = 5.10), confirming the robustness of our statistical approach.

After addressing concerns about extreme odds ratios through variable standardization, the analysis revealed that average similarity to others shows a moderate positive effect (OR = 1.35 per SD). This indicates that ideas semantically similar to existing ones are more likely to be selected, a pattern that reflects human preference for comprehensible and contextually relevant ideas.

Furthermore, the contrasting effects of maximum similarity (OR = 1.41 per SD) and minimum similarity (OR = 0.824 per SD) indicate that ideas likely to be selected need to satisfy balanced similarity conditions. The strong negative effect of similarity standard deviation (OR = 0.407 per SD) suggests that ideas with extreme variability in similarity are less likely to be selected.

A significant negative positional effect was confirmed for grid row (OR = 0.719 per SD) and grid column (OR = 0.664 per SD). This means that the closer an idea is to the top-left of the grid, the higher its selection rate, indicating that visual placement systematically affects idea evaluation due to UI design constraints. As visualized in Fig 7, the heatmap clearly demonstrates this positional bias, with the highest selection rates concentrated in the upper-left regions of the grid and progressively decreasing toward the bottom-right areas.

This bias represents a fundamental limitation of grid-based creative interfaces that future research must address.

Additional analysis controlling for idea creator (AI vs. human) revealed no significant creator effect (OR = 1.01, p = 0.945), indicating that selection patterns are driven by idea characteristics rather than their origin. Importantly, while AI strategies do influence position distribution (Vertical strategy concentrates in specific columns through idea development, while Horizontal strategy explores new columns), the analysis revealed that strategy effects extend beyond mere positional influence. The t-SNE visualization in Fig 8 provides additional evidence for distinct strategy-based clustering patterns, where ideas generated under different AI strategies occupy different regions in the semantic space, indicating that the strategies produce qualitatively different types of content beyond spatial positioning. After controlling for position

**Table 7. Complete strategy pairwise comparisons.** Mixed-effects logistic regression results showing all pairwise comparisons between AI strategies with Creator control variable (N = 5,848 ideas from 148 participants).

| Comparison | Odds Ratio | 95% CI | p-value |
|---|---|---|---|
| Vertical vs. Random | 1.13 | 0.658-1.95 | 0.658 |
| Horizontal vs. Random | 1.04 | 0.600-1.81 | 0.882 |
| Vertical vs. Horizontal | 1.09 | 0.626-1.89 | 0.765 |

Note: All strategy comparisons are non-significant when controlling for Creator effects and examining all ideas collectively. Effect sizes are small (< 3 percentage points difference in predicted probabilities), indicating limited practical significance. Analysis addresses reviewer concerns about incomplete strategy comparisons.

effects in the mixed-effects model, content-based factors such as average similarity (OR = 1.35 per SD) remained highly significant, demonstrating the robustness of content-related selection mechanisms.

This result suggests cognitive constraints in the human idea evaluation process. Extremely novel ideas (the negative effect of minimum similarity) are less likely to be selected due to difficulty in comprehension, whereas ideas that can be connected to the existing context (the strong positive effect of average similarity) are easier to evaluate due to lower cognitive load.

The coexistence of positional and content effects indicates that idea evaluation involves both presentation-dependent and content-dependent mechanisms. Crucially, interaction analysis showed no significant strategy×position interactions, confirming that positional bias affects all strategies equally and does not confound strategy comparisons.

### Convergent strategies for divergent goals

Integrating findings from RQ1-RQ3 and additional robustness analyses provides a coherent understanding of how AI strategies influence collaborative creativity. RQ1 showed that AI strategies clearly impact human trust, and RQ2 demonstrated significant effects specifically on the adoption of AI-generated ideas. RQ3, conversely, revealed the fundamental mechanisms governing idea selection across all contributions, showing these mechanisms function consistently regardless of idea origin (AI vs. human) or AI strategy. The absence of significant main effects for AI strategies in RQ3's comprehensive analysis is not contradictory; rather, it suggests AI strategies achieve their specific effectiveness (as seen in RQ2) by generating ideas that align with universal human selection preferences, rather than by fundamentally altering the selection criteria themselves.

With careful attention to different analytical perspectives, this study's most significant theoretical implication is the phenomenon that "from the perspective of human selection as an evaluation metric, a convergent strategy (deepening) is effective in brainstorming, which has a divergent goal."

Our findings challenge the prevailing assumption in human-AI creativity research that emphasizes diversity and exploration. Previous work, like Liu et al.'s [33] CoQuest system, demonstrated clear advantages for breadth-first (exploration) strategies in research question generation. In CoQuest, participants preferred this approach because it was easier to compare and explore multiple research questions. However, our results suggest these advantages may not generalize to turn-based collaborative ideation. The critical distinction lies in the interaction model and evaluation criteria: CoQuest's batch generation positioned AI as an advanced search tool, where diversity directly enhanced user selection and cognitive control. In contrast, our turn-taking conversational model positioned AI as an equal dialogue partner, where predictability and relevance became paramount for maintaining collaborative trust and engagement.

As revealed in RQ3, the positive effect of average similarity (OR = 1.33 per SD) indicates a consistent human tendency to select ideas relevant to existing ones. This contrasts with traditional brainstorming theory, such as Osborn's [18] "quantity breeds quality," and suggests human-AI collaboration operates under different cognitive and social dynamics than human-only brainstorming or tool-based AI assistance. The effectiveness of the deepening strategy (confirmed in RQ2) is thus understood in this context; by incrementally developing ideas from human input, the AI naturally maintains relevance. Conversely, entirely isolated ideas are less likely to be selected (negative effect of minimum similarity: OR = 0.824 per SD). Consequently, ideas from the deepening strategy align well with human selection patterns in collaborative settings.

This finding points to a new mechanism in human-AI collaboration that deviates from both traditional creativity theory and recent AI creativity research. In effective collaboration, when AI prioritizes "acceptable development" over "diversity expansion," the actual creative outcome (selected ideas) improves. However, it's crucial to note that this study measured "selectability by humans," and whether selected ideas are truly creative requires separate evaluation. This suggests a unique creative process specific to human-AI collaboration that transcends the traditional dichotomy of divergence and convergence, challenging the assumption that AI should primarily serve as a generator of diverse alternatives.

 

Considering the increase in trust confirmed in RQ1, the analysis of the similarity effect supports the idea that in human-AI collaboration, AI functions not as a "catalyst for divergence" but as a "promoter of development." By providing new developments while maintaining relevance to initial human ideas, AI can make creative contributions consistent with human selection behavior while building the trust necessary for sustained collaboration.

Beyond immediate outcomes, these patterns may also have implications for the kinds of collaboration-relevant competencies that could be fostered through repeated human–AI interaction. In particular, the predictability and incremental nature of the deepening strategy—together with its higher trust ratings and greater adoption of AI ideas—may provide conditions under which users can gradually learn to engage with AI contributions more effectively over time. While we did not directly measure skill acquisition in this study, we note that this possibility is consistent with broader discussions of competency development through sustained interaction with workplace technologies and learning-by-doing in human–technology interaction [56]. This motivates future work that directly tests whether such learning occurs and whether it transfers beyond the collaborative context.

## Practical implications

The empirical results of this study provide concrete guidelines for the design of human-AI creative collaboration systems. The superiority of the deepening strategy, confirmed by the experiment, can be systematized into three design principles. First, the deepening strategy is effective when prioritizing trust-building in an AI collaboration system. The deepening strategy showed significant improvements in both Competence Trust ($F(2, 145) = 5.07, p = 0.007$) and Moral Trust ($F(2, 145) = 4.12, p = 0.018$). This effect is likely due to the clarity of the strategy, as its recognition rate was high at 61.5%, compared to the 28.0% unclear rate for the broadening strategy. Second, when prioritizing the actual adoption rate of AI ideas, a 35% higher selection rate can be expected with the deepening strategy. This demonstrates that incremental idea development enhances receptivity. However, an important caveat here is that the "selection rate" measured in this study reflects immediate human receptivity and may not necessarily align with long-term creative value or true innovativeness. Third, a relationship is suggested between the participant's understanding of the AI's behavior pattern and collaborative effectiveness. Although 61.5% of participants in the deepening strategy accurately recognized the strategy and simultaneously showed superiority in trust and idea selection rates, the causal relationship between recognition and effectiveness requires further investigation.

While the experiment involved a static comparison of strategies, the results allow for the derivation of implementation guidelines for dynamic strategy adjustment. It would be effective to implement a function where the system monitors participant behavior indicators (e.g., AI idea selection rate, session continuation intent) in real-time and considers switching to the deepening strategy if a decline in collaborative effectiveness is detected. Moreover, the discovery of the grid effect offers important implications for visual interface design. As demonstrated in Fig 7, the clear positional bias toward the upper-left regions suggests that it would be effective to place important AI ideas in the area users see first and to promote strategy understanding through the visualization of incremental development. However, it should be noted that this study did not verify the necessity of scrolling or aspects of accessibility, and the effect might simply be due to visually accessible positions. Furthermore, by leveraging the similarity effect identified by the mixed-effects logistic regression analysis (*avg similarity to others*, OR = 1.33 per SD), the adoption rate can be optimized by appropriately controlling the average similarity of AI-generated ideas to existing ones.

While these principles may apply broadly, three sectors are particularly likely to benefit from our findings. First, marketing and advertising professionals can leverage the deepening strategy for campaign ideation—our experimental task (increasing café sales) directly reflects this domain. The strategy's predictability enables transparent AI integration, and the number of selected AI ideas increased by 34% compared to the broadening strategy.Second, product development teams may benefit from applying the incremental approach to bridge the gap between ideation and implementation. The deepening strategy generates ideas that build upon existing concepts, which our findings suggest are more likely to be

adopted (34% increase in selected AI ideas) and trusted (significant improvements in Competence Trust and Moral Trust). In product development contexts where ideas must eventually be implemented, this characteristic may help produce concepts that remain grounded in practical constraints rather than diverging into infeasible directions.Third, educational institutions developing AI literacy curricula may leverage a key finding from this study: 61.5% of participants accurately recognized the deepening strategy's behavioral pattern. This high recognition rate suggests that the strategy's incremental, predictable nature could make AI behavior observable and learnable. Educators might use this transparency to help students understand how AI contributes to collaborative work, potentially supporting the development of skills for effective human-AI teaming.

For adoption across these sectors, we recommend initially deploying deepening-oriented systems to establish trust, with potential introduction of broadening strategies once familiarity is established.

## Limitations and future work

This study has several important limitations. First, regarding experimental design, the validation was limited to a single short session, and changes in strategy effectiveness in long-term collaborative relationships were not examined. Trust in long-term AI–human collaboration can vary over time [13], so a longer-term perspective beyond a single session is needed. The experiment was also limited to a specific domain, "increasing cafe sales," and the effectiveness of the strategies in other creative tasks such as technical problem-solving or artistic creation remains unverified. Therefore, caution is warranted when generalizing to other domains; given empirical arguments that creativity is at least partly domain-specific [57], whether the same support strategies are effective in other domains requires future examination. Furthermore, the qualitative evaluation of the ideas selected by participants was not conducted by experts, and evaluating strategy effectiveness solely based on selection rate has the limitation of overlooking the inspirational effect of unelected ideas. As an important theoretical limitation, this study used "selectability by humans" as its primary metric, and whether this aligns with true creativity or long-term innovation value remains unverified. The "easily accepted" ideas generated by the deepening strategy are not necessarily the most creative or innovative ideas. Additionally, the study was conducted only with Japanese participants, so the influence of cultural factors is unknown. Because cultural factors can shape norms for participation, evaluation, and novelty preferences in co-creative work [58], cross-cultural validation of our AI–human brainstorming strategy will be essential.

Second, regarding technical and statistical limitations, the results of this study are based on the capability level of GPT-4.1, and the patterns of strategy effectiveness may change with the rapid development of AI technology. The visual constraints of the 15x15 grid created a strong positional bias toward the top-left area (OR = 0.719 for rows, OR = 0.664 for columns after standardization). While we controlled for positional variables in the mixed-effects model and confirmed no significant strategy×position interactions (all p >0.05), the compound effects of UI design and strategy implementation represent an inherent limitation. Notably, however, content-based effects remained significant after controlling for position, indicating that strategies have substantive impacts beyond spatial positioning. Additional robustness analyses revealed minor multicollinearity concerns (one variable with VIF = 5.10) and confirmed that AI strategy effects, while significant for AI-specific idea adoption (RQ2), showed no significant main effects when examining all ideas collectively (RQ3). The AI's strategy implementation was rule-based, and the adaptive strategy adjustment required in actual AI collaboration was not realized. Furthermore, the dynamic changes in participants' cognitive processes and trust formation were evaluated only through post-session questionnaires.

Based on these limitations, the following are suggested as future research topics. The development of adaptive strategy systems requires the evaluation of systems that dynamically adjust strategies according to the user's state. The expansion of measurement requires the elucidation of cognitive processes through protocol analysis, multi-faceted evaluation of ideas by experts, and dynamic trust measurement. Particularly important is the evaluation of the long-term creative value of selected ideas and the investigation of how selection bias affects the quality of creativity.The expansion of

experimental conditions requires investigation of trust changes in long-term collaboration, validation in different creative tasks, and cross-cultural comparisons. For theoretical development, the construction of a new framework to explain the unique dynamics of human-AI collaboration is necessary. Regarding the positional effect, a more rigorous verification is required through sensitivity analysis or randomization of initial placements.

Despite these limitations, this study provides a significant theoretical and practical contribution by establishing fundamental guidelines for interaction design in human-AI creative collaboration with empirical data.

## Conclusion

To address the problem of interaction design in human-AI creative collaboration, this study examined whether an AI should prioritize the "deepening" or "diversification" of human thought. This research serves as an example of how to identify effective AI collaborative strategies. The experiment was conducted with a between-participants design with three conditions, where 148 participants engaged in brainstorming under a deepening strategy (Vertical), a broadening strategy (Horizontal), or a random control condition. The dependent variables were trust in the AI and idea selection rate.

The results showed that the deepening strategy was significantly superior to the broadening strategy in building trust , and the selection rate of AI ideas was also higher . Additionally, predictive probability analysis revealed a 2.36 percentage point improvement for the Vertical strategy over Random, demonstrating substantial practical significance. Furthermore, 61.5% of participants accurately recognized the deepening strategy, indicating that the clarity of the strategy is important for collaborative effectiveness. The mixed-effects logistic regression analysis confirmed a pattern different from conventional brainstorming theory, where ideas with balanced similarity to existing ideas were more likely to be selected (OR = 1.33 per SD after standardization).

These results support our hypotheses. It was also found that collaborative effectiveness is influenced by the type of strategy and its understandability. This study is an important example of interaction design in human-AI creative collaboration. It was found that AI collaboration systems, which are increasingly used in human society, can gain trust and acceptance from humans through a deepening strategy. Future research can develop adaptive AI collaboration systems that consider individual differences and long-term collaboration, and that can be applied to various creative tasks.

## Supporting information

**S1 File. AI strategy prompt templates and technical implementation.** Complete prompt templates used for implementing the Vertical (deepening) and Horizontal (broadening) AI strategies, including task definitions, contextual information requirements, and behavioral constraints for each strategy condition. Also includes technical implementation details: OpenAI API configuration with GPT-4.1, temperature=0.0, max_tokens=400, n=1, and response_format configured as JSON object. Contains detailed template variables, design rationale, and technical configuration parameters used across all experimental conditions.
(DOCX)

**S2 File. User Interface Instructions and Experimental Materials.** Comprehensive user instructions for the frontend interface, including video explanations, screen-by-screen interface descriptions, and complete questionnaire items. Contains detailed instructions for brainstorming session operations, AI behavior evaluation scales (GQS and MDMT), and all experimental materials used for participant guidance throughout the study.
(DOCX)

**S3 File. Complete dataset.**
(XLSX)

## Author contributions

**Conceptualization:** Kazuki Komura, Seiji Yamada.

**Data curation:** Kazuki Komura.

**Formal analysis:** Kazuki Komura.

**Funding acquisition:** Kazuki Komura, Seiji Yamada.

**Investigation:** Kazuki Komura.

**Methodology:** Kazuki Komura, Seiji Yamada.

**Project administration:** Kazuki Komura.

**Resources:** Kazuki Komura.

**Software:** Kazuki Komura.

**Supervision:** Kazuki Komura, Seiji Yamada.

**Validation:** Kazuki Komura.

**Visualization:** Kazuki Komura.

**Writing – original draft:** Kazuki Komura.

**Writing – review & editing:** Kazuki Komura.

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
