## [Decision Letter · Decision Letter 0]

1 Sep 2025

PONE-D-25-40481Deepening Ideas vs. Exploring New Ones: AI Strategy Effects in Human-AI Creative CollaborationPLOS ONE

Dear Dr. Komura,

Thank you for submitting your manuscript to PLOS ONE. After careful consideration, we feel that it has merit but does not fully meet PLOS ONE’s publication criteria as it currently stands. Therefore, we invite you to submit a revised version of the manuscript that addresses the points raised during the review process.

We look forward to receiving your revised manuscript.

Kind regards,

Md. Rabiul Awal

Academic Editor

PLOS ONE

Assistant Professor in Management

Bangladesh army University of science and Technology

[JST, CREST (JPMJCR21D4), Japan.].

Additional Editor Comments (if provided):

Reviewer #1:

Reviewer #2:

Reviewers' comments:

Reviewer's Responses to Questions

**Comments to the Author**

1. Is the manuscript technically sound, and do the data support the conclusions?

Reviewer #1: Yes

Reviewer #2: Yes

2. Has the statistical analysis been performed appropriately and rigorously?

Reviewer #1: Yes

Reviewer #2: Yes

3. Have the authors made all data underlying the findings in their manuscript fully available?

Reviewer #1: Yes

Reviewer #2: Yes

4. Is the manuscript presented in an intelligible fashion and written in standard English?

Reviewer #1: Yes

Reviewer #2: Yes

5. Review Comments to the Author

Reviewer #1: The scope of the study is limited. It only focuses on a specific task (brainstorming for increasing cafe sales), which might not generalize to other creative tasks or contexts. No tests for the implications for varied domains

No long-term effects are addressed; only the experiment is conducted in a single session, limiting insights into the long-term dynamics of human-AI collaboration. Long-term effects and the evolution of trust over time are not addressed.

Only homogeneous culture is discussed; no cross-cultural differences are discussed

Reviewer #2: The study investigates human-AI co-creativity and how an AI like GPT-4.1 can play a supportive role in human idea generation. The authors compare two AI strategies: exploration (finding new ideas) and exploitation (deepening of ideas) and analyze which strategy leads to higher trust in the AI-collaborative partner and higher idea adoption.

The choice of the exploration-exploitation theory as a theoretical foundation is very well suited for this research and positioned neatly in relation to creative ideation. The authors further refer to highly relevant theories in creativity research (Guilford, Mednick, Finke) and build a deepened framework for the study at hand.

The methods chosen are very well suited for answering the research questions. Especially the choice of putting new ideas on a visual grit horizontally (indicating new idea exploration) or vertically (indicating deepening of an old idea) allows the participants to approach the task in an intuitive approach.

Furthermore, the research questions are well formed to explore the research interests, and the limitations and future avenues for research are discussed critically and insightfully.

I only have a few comments that I hope the authors can clear up in a revised version of their manuscript.

Comments:

Line 270: “Each cell in the 15×15 grid space can contain a maximum of one idea (in a sticky note format). This constraint is intended to encourage participants to engage in strategic spatial placement and stimulate creative thinking”.

Participants are encouraged to do so, but were they also explicitly instructed to do strategic planning? Were they given examples and able to get familiar with the design through a mock example?

Lines 280-281: “In the experiment, the human and the AI each generated 20 ideas, creating a total of up to 40 ideas.”

Did they have to come up with exactly 20 ideas each? Or if they ran out of ideas before that they could finish early? Was there a time limit to how long they were allowed to think for each idea? It sounds as if they were required to write exactly 20 ideas (as it is also mentioned that their progress is tracked on a progress bar), but then saying “a total of up to 40 ideas” sounds as if it could be less than 40 ideas. Same at line 383: “maximum total of 40 ideas” which sounds as if it could be less.

Line 309: In the deepening-ideas condition it says “the parent idea to be deepened”. Does the parent idea only refer to the human original idea? Or can the AI also deepen a response that it gave by itself in response to the human original idea? If the human participant wrote a new idea in response to the AI deepened idea, will the AI deepen the first original human idea further or switch to the new human idea and deepen or expand this further?

Also, it is a valid choice to stop deepening an idea after 9 instances of an expansion - but does the limit of 9 refer to the number of answers the AI can give or in total? So if the humans themselves have already written 5 ideas in that specific column, will the AI only deepen this category 4 more times, or another 9 times?

Line 543: “This study empirically validated two AI strategies”. Can you really say that the study “validated” these two strategies? Validation means that both strategies were proven as being correct, but the study at hand rather assesses and compares both strategies. “Validated” carries a very strong connotation of correctness which is not the research question here.

6. PLOS authors have the option to publish the peer review history of their article (what does this mean?). If published, this will include your full peer review and any attached files.

Reviewer #1: No

Reviewer #2: No

---

## [Author Response · Author response to Decision Letter 1]

12 Oct 2025

Dear Editor and Reviewers,

Thank you very much for your thoughtful and constructive feedback and for the opportunity to revise our manuscript. Please see the attached “Response to Reviewers” for point-by-point replies. We have uploaded the revised manuscript (with changes), the clean manuscript, and updated figures.

We sincerely appreciate your time and consideration.

Sincerely,

Kazuki Komura

---

## [Decision Letter · Decision Letter 1]

2 Dec 2025

PONE-D-25-40481R1Deepening Ideas vs. Exploring New Ones: AI Strategy Effects in Human-AI Creative CollaborationPLOS ONE

Dear Dr. Komura,

Thank you for submitting your manuscript to PLOS ONE. After careful consideration, we feel that it has merit but does not fully meet PLOS ONE’s publication criteria as it currently stands. Therefore, we invite you to submit a revised version of the manuscript that addresses the points raised during the review process. 

1. The manuscript presents interesting and timely insights into human–AI collaboration; however, to further strengthen the contribution, the authors should more explicitly address the role of human–AI collaboration in the development of human competencies. This aspect carries both theoretical and practical significance and would enrich the discussion on how AI can support—not only complement—human creative and cognitive capabilities. I recommend that the authors incorporate relevant perspectives from recent research on competence development in human–technology interactions . The following article offers a useful conceptual foundation https://doi.org/10.1108/JOCM-10-2023-0426. Integrating this dimension would enhance the theoretical depth of the manuscript and clarify the practical implications of designing AI systems that not only collaborate with humans but also help them learn, grow, and expand their capabilities. Please note that citation of this is entirely optional but we hope it will be helpful overall.

2. In the Practical Implications section, you stated that “the empirical results of this study provide concrete guidelines for the design of human–AI creative collaboration systems.” However, the manuscript does not offer specific, actionable guidance for different types of organizations. Please clarify which industries, sectors, or organizational contexts could directly benefit from your findings

We look forward to receiving your revised manuscript.

Kind regards,

Katarzyna Piwowar-Sulej

Academic Editor

PLOS ONE

Journal Requirements:

Reviewers' comments:

Reviewer's Responses to Questions

**Comments to the Author**

1. If the authors have adequately addressed your comments raised in a previous round of review and you feel that this manuscript is now acceptable for publication, you may indicate that here to bypass the “Comments to the Author” section, enter your conflict of interest statement in the “Confidential to Editor” section, and submit your "Accept" recommendation.

Reviewer #2: All comments have been addressed

2. Is the manuscript technically sound, and do the data support the conclusions?

Reviewer #2: Yes

3. Has the statistical analysis been performed appropriately and rigorously?

Reviewer #2: Yes

4. Have the authors made all data underlying the findings in their manuscript fully available?

Reviewer #2: Yes

5. Is the manuscript presented in an intelligible fashion and written in standard English?

Reviewer #2: Yes

6. Review Comments to the Author

Reviewer #2: The authors have included all my suggestions and added more explanations about the experimental procedure and study design.

With this the manuscript has improved significantly. I consider it now fit for publication.

7. PLOS authors have the option to publish the peer review history of their article (what does this mean?). If published, this will include your full peer review and any attached files.

Reviewer #2: No

---

## [Author Response · Author response to Decision Letter 2]

14 Dec 2025

Response to Academic Editor

We sincerely thank the editor for the time, careful reading, and constructive comments. We have revised the manuscript accordingly and indicate specific changes and locations below.

“The manuscript presents interesting and timely insights into human–AI collaboration; however, to further strengthen the contribution, the authors should more explicitly address the role of human–AI collaboration in the development of human competencies. This aspect carries both theoretical and practical significance and would enrich the discussion on how AI can support—not only complement—human creative and cognitive capabilities. I recommend that the authors incorporate relevant perspectives from recent research on competence development in human–technology interactions . The following article offers a useful conceptual foundation https://doi.org/10.1108/JOCM-10-2023-0426. Integrating this dimension would enhance the theoretical depth of the manuscript and clarify the practical implications of designing AI systems that not only collaborate with humans but also help them learn, grow, and expand their capabilities. Please note that citation of this is entirely optional but we hope it will be helpful overall.”

We agree that this dimension would strengthen the contribution. To address this point while keeping our claims appropriately cautious (since our study does not directly measure skill acquisition), we added a concise discussion that explicitly connects our empirical findings to the possibility that certain forms of human–AI interaction may support learning and competency development over time.

Concretely, we added a paragraph at the end of the “Convergent Strategies for Divergent Goals” subsection in the Discussion (clean manuscript: p.19, lines 697–707) to (i) link our empirical findings to the possibility that repeated human–AI interaction in creative tasks may foster collaboration-relevant competencies over time (e.g., learning to engage with AI contributions more effectively as users become familiar with the AI’s interaction pattern), (ii) connect this perspective to our observed results (e.g., higher trust ratings and greater adoption of AI ideas under the deepening strategy), and (iii) clearly state the limitation that competency development was not directly measured and should be examined in future longitudinal work. In this added text, we cite the suggested work as a conceptual foundation for competency development in human–technology interactions (added as Ref. [56]).

“In the Practical Implications section, you stated that “the empirical results of this study provide concrete guidelines for the design of human–AI creative collaboration systems.” However, the manuscript does not offer specific, actionable guidance for different types of organizations. Please clarify which industries, sectors, or organizational contexts could directly benefit from your findings”

We thank the Academic Editor for this important point. We expanded the Practical Implications section to provide specific, actionable guidance for organizational contexts that could directly benefit from our findings (clean manuscript: p.20, lines 742–763). Specifically, we now identify representative sectors/organizational settings (e.g., marketing/advertising ideation, product development/innovation teams, and educational settings for AI literacy and creative thinking) and explain how the observed strengths of the deepening strategy translate into practical design and deployment guidance. We also adopt a cautious tone to avoid overgeneralization beyond our experimental setting.

In addition, we make the added guidance explicit by summarizing the three design principles (trust-building, short-term adoption of AI ideas, and the link between understanding AI behavior and collaboration effectiveness) and by describing implementation guidance such as starting with deepening-oriented systems to establish trust before introducing broadening strategies once familiarity is established.

Response to Reviewer #2

We sincerely thank Reviewer #2 for their positive evaluation and for confirming that all prior comments have been addressed. No additional changes were requested by Reviewer #2 in this round, and our revisions focused on the Academic Editor’s two points above.

---

## [Editor Report · Decision Letter 2]

23 Dec 2025

Deepening Ideas vs. Exploring New Ones: AI Strategy Effects in Human-AI Creative Collaboration

PONE-D-25-40481R2

Dear Dr. Komura,

We’re pleased to inform you that your manuscript has been judged scientifically suitable for publication and will be formally accepted for publication once it meets all outstanding technical requirements.

Kind regards,

Katarzyna Piwowar-Sulej

Academic Editor

PLOS One

---

## [Editor Report · Acceptance letter]

PONE-D-25-40481R2

PLOS One

Dear Dr. Komura,

I'm pleased to inform you that your manuscript has been deemed suitable for publication in PLOS One. Congratulations! Your manuscript is now being handed over to our production team.

Kind regards,

on behalf of

Professor Katarzyna Piwowar-Sulej

Academic Editor

PLOS One